# A Critical Review of Electroporation as A Plasmid Delivery System in Mouse Skeletal Muscle

**DOI:** 10.3390/ijms20112776

**Published:** 2019-06-06

**Authors:** Emilia Sokołowska, Agnieszka Urszula Błachnio-Zabielska

**Affiliations:** Department of Hygiene, Epidemiology and Metabolic Disorders, Medical University of Bialystok, 15-222 Bialystok, Poland; agnieszka.blachnio@umb.edu.pl

**Keywords:** electroporation, plasmids, non-viral, animal models, mouse, gene electrotransfer, muscle, silencing

## Abstract

The gene delivery to skeletal muscles is a promising strategy for the treatment of both muscular disorders (by silencing or overexpression of specific gene) and systemic secretion of therapeutic proteins. The use of a physical method like electroporation with plate or needle electrodes facilitates long-lasting gene silencing in situ. It has been reported that electroporation enhances the expression of the naked DNA gene in the skeletal muscle up to 100 times and decreases the changeability of the intramuscular expression. Coelectransfer of reporter genes such as green fluorescent protein (GFP), luciferase or beta-galactosidase allows the observation of correctly performed silencing in the muscles. Appropriate selection of plasmid injection volume and concentration, as well as electrotransfer parameters, such as the voltage, the length and the number of electrical pulses do not cause long-term damage to myocytes. In this review, we summarized the electroporation methodology as well as the procedure of electrotransfer to the gastrocnemius, tibialis, soleus and foot muscles and compare their advantages and disadvantages.

## 1. Introduction

Gene therapy is becoming a promising tool for innovating treatment for dangerous diseases [1]. Many of these conditions require the systemic delivery of a therapeutic protein. It seems that injection of naked plasmid DNA offered advantages like a reduction of toxicity, safety [2] and lead to physiological and therapeutic responses. Plasmids are theoretically an excellent material for constructing gene therapy tools and vaccines [3] delivered into specific organs and tissues, including skeletal muscle, where promote strong humoral and cellular immune responses. There is a lot of interest in supplying genes to skeletal muscle tissue, especially in such disease entities as Duchenne muscular dystrophy [4]. However, this process is problematic, because of ineffective plasmid capture by muscle fibres. Despite the intramuscular injection of a large volume of plasmid most of them are probably left in the extracellular space and it is hydrolysed by DNAases [5,6]. It turns out that the expression of plasmid DNA in the muscles can be prolonged [7,8] and dramatically increased especially when the electric current is applied. Plasmid DNA should be delivered to the tissue placed between the electrodes in which the electric field is used [9,10]. 

This type of gene therapy seems to be excellent for the local secretion of angiogenic or neurotrophic factors, as well as for efficient delivery of other large molecules, such as oligonucleotides [11]. Another interesting application is the use of well-vascularized muscles as an endocrine organ that provides systemic therapeutic proteins, such as erythropoietin, coagulation factors, anti-inflammatory cytokines [5,12,13,14,15]. In addition to skeletal muscle tissue, electrotransfer was successfully performed in the whole embryos [16], liver [17], kidneys [18], joints [19], brain [20], cornea [21], lungs [22], myocardium [23], carotid artery [24], retina [25], the skin [26] and sperm [27]. 

Since the pioneering publication by Neumann et al. in 1982, the in vivo electropermeabilization (also termed electroporation) emerged as a promising tool for the transfer of exogenous drugs and nucleic acids [28] because of its low cost, its safety in production and utilization and the lack of risk of insertional mutagenesis [29]. The authors [28] have used the most favourable parameters of electrical pulses to obtain the desired levels of cell permeabilization and deliver (normally impermeable) molecules such as DNA into cells [30]. Suitable electrical pulses produce small gaps in the cell membrane through which plasmids can enter the cytoplasm of the cell and next into the nuclei where the transgene is expressed. Then, after the plasmids enter the cells, the cell membrane branches get rapidly closed [1,31]. This gaps in the cell membrane pose a risk of necrosis of some muscle fibres, called collateral damage [32]. A vast majority of experiments conducted over the last three decades focus on optimizing the electrical impulse protocol and other parameters, like the use of hyaluronidase, estimation of the interval between the injection of plasmid DNA and the delivery of electrical pulses, electrode geometry, tissue properties (the strain and age of used animals), that were thought to affect the efficiency and safety of skeletal muscle electrotransfection [33,34,35,36,37]. However, the results of studies assessing the most optimal electrotransfer parameters are very diverse among different authors and do not present an unambiguous, appropriately selected protocol for the most effective electrotransfection of skeletal muscles. In this review, based on the literature and experiences of our research team, we discussed the parameters of electroporation of gastrocnemius muscles, tibialis anterior, soleus and foot muscles in mice that limit muscle damage and are crucial for accurate physiological, biochemical and molecular measurements.

## 2. Electroporation

The selection of an appropriate method of introducing exogenous genetic material (transfection) into eukaryotic cells is a complex process and depends on many factors. Electroporation allows monitoring of the spread of labelled DNA in the cell [38,39]. The use of exogenous plasmid DNA (pDNA) in gene therapy requires the expression of a gene cloned on a plasmid. The most beneficial is the delivery and accumulation of DNA in the cell nucleus. However, this is not easy to achieve due to the slow penetration of naked DNA into the cell. Additionally, endolysosomal compartments can intercept and degrade genetic material [40]. Plasmid DNA has to overcome three main barriers: cell membrane, cytoplasm and nuclear membrane [41]. In the case of electrotransfer of plasmids to muscle tissue, the extracellular matrix layer (ECM) is an additional obstacle, which can be divided into epimysium (surrounding muscle), perimysium (surrounding muscle bundles) and endomysium (surrounding muscle fibres) [42] (Figure 1). Exogenous pDNA introduced into the cell can maintain almost unchanged mobility but this applies only to small molecules (in the range of 500–750 Da). Movement in the cytoplasm rapidly decreases with the increased molecule size, therefore, the mobility of free plasmids in the cytoplasm almost does not exist [43]. Hence, the plasmid DNA is actively carried through the cytoplasm by microtubules (fast transport) [44] and along actin filaments [39] (slow transport). Recently, interactions between pDNA introduced into the cell by electroporation and dyneins (motor proteins) have been demonstrated [45]. The detailed mechanism of the electrotransfer of DNA molecules into cells through the cell membrane and further passage through the cytoplasm into the cell nucleus has been described in detail in the excellent article by M. P. Rols [46].

Electropermeabilization, commonly known as electroporation, is a physical procedure that engages the induction of a short, intense electric field that creates brief branches to the plasma membrane. Such damage to the membrane allows the inflow of large particles into the cell (Figure 1). This technique has been used for nearly three decades for the transfection of cells in vitro [47] and next to in vivo transfer DNA into the skin [1], tumours [48] and hepatocytes [49]. In 1998, Aihara and Miyazaki were the first to transfer DNA into muscle fibres [8]. Currently, electroporation is used both in the treatment of muscular dystrophy but also for the long-term systemic delivery of therapeutic proteins through muscles, which in this case serve as an endocrine organ [7,50,51,52,53]. Electroporation is also an effective delivery method for improving immune responses elicited by DNA vaccines. Electroporation by itself is able to elicit humoral and cellular immune responses, hence the idea that electroporation has adjuvant-like properties in vaccination when coupled with plasmid DNA injection [54,55,56]. The mechanism of electroporation, which increases the expression of the transgene by applying an electric current, seems to be rather uncomplicated. Even though the tools, requirements, methods and animal models vary extensively, in all procedures, it is relevant to apply electric current only after injecting plasmid DNA into the muscle. The electric pulse enhances the membrane permeability but primarily acts directly on the charged DNA molecules, thanks to which the plasmids migrate through the cellular membranes to the cytoplasm [7]. 

The use of appropriate electrical pulses is one of the most relevant parameters of electrotransfection. An eccentric field applied to the muscle should provide permeabilization of the cell membrane but cannot be too high to avoid cells irreversible damage. Different protocols evaluated properly selected square wave electric pulses for the efficient electrotransfusion of mouse skeletal muscle, including the delivery of low voltage electrical pulses and the combination of high and low voltage electrical current. Initially, only low voltage (LV) electrical pulses with a long duration were used (100–200 V/cm, 20–50 ms) [7,8,57]. Subsequent studies achieved effective transfection by using high voltage (HV) pulses where the voltage parameters are above 400 V/cm and are expressed in microseconds (usually 50–100 μs) followed by a different number of long electrical pulses with a voltage below 400 V/cm which are expressed in milliseconds (usually 10–400 ms) [58,59,60].

## 3. The Usefulness and Safety of Plasmid Vectors for Electroporation

Adenovirus vectors, retroviral vectors and plasmids are the most commonly used gene delivery carriers in clinical trials of new gene therapies [61] (Table 1). 

It is possible to deliver transgenes to skeletal muscles for the therapeutic gene reinstatement by using viral or non-viral vectors [62,63]. Growing attention is paid to the supply of genes using plasmid carriers, mainly due to their lower toxicity and ease of production [64]. However, the effectiveness of plasmid-based gene delivery to muscle fibres by intramuscular injection still does not provide an efficient transfection [65,66]. Probably, plasmids can reach muscle fibres only by non-specific vacuole formation in the process of endocytosis due to the lack of typical receptors or carriers for DNA on the surface of the sarcolemma [5]. Nevertheless, animal experiments demonstrate that the use of electric current during electroporation [7,8,33,34], or ultrasound in the sonoporation [67,68,69], significantly increases the efficiency of plasmid-based gene transfer (Table 2). The size of the plasmid is also relevant. Larger plasmids form clusters that are less likely to pass through the membrane branches created during electroporation. 

Afterward, using plasmid DNA in gene therapy poses numerous advantages. First, retroviruses vectors can cause insertional mutations during the integration process into genomic DNA [70]. Secondly, the possibility exists of reproduction of replicable viral particles when viral origin vectors are used [71]. Plasmids as non-viral carriers provide high insert capacity, the lack of host immune response to the plasmid, the absence of non-immunological toxicity and the profitability of its manufacture on a large, high-quality scale. Plasmid vectors do not encode other antigens than the transgene product itself. That gives an advantage over the viral vectors against which an immune response can be directed, especially in the case of the need for further administrations. Additionally, vectors derived from common human pathogens may encounter antibodies induced by previous infections, which may interfere with the therapy itself [72]. Naked plasmid DNA has low immunogenicity and can be repeatedly administered, although local inflammatory reactions are still a problem [73].

An important issue is the degree of purity of the pDNA. Intramuscular administration of some pDNA preparations in mice resulted in muscle necrosis. In turn, other DNA preparations did not cause any changes. This phenomenon was not related to the plasmid sequence, its size or encoded genes but with contamination with fragments of bacterial genomic DNA [74,75].

## 4. Markers of Electroporation: Luciferase, A Green Fluorescent Protein (GFP), Beta-Galactosidase

Reporter genes are widely used in pharmacological and biomedical research, in molecular biology and biochemistry, as indicators of gene expression. Typically, the reporter gene and the gene of interest are placed in the same DNA construct, which is then delivered into the selected cell or organism. In most experiments luciferase, a green fluorescent protein or beta-galactosidase serve as a positive control. Bioluminescence imaging uses CCD (charge coupled device) cameras, which are very sensitive and allow to determine both the number and exact location of photons in the sample or the body [76]. The higher efficiency of the CCD camera can be achieved by its cooling to about −80 ° C before the imaging starts, which reduces the occurrence of thermal noise [77]. 

Luciferase as a reporter gene is widely used in electroporation, due to its sensitivity of detection and the ease of quantification compared to other reporter genes facilitating the analysis of the transcriptional activity. Luciferase reporter technology is based on the interaction of the luciferase enzyme with a luminescent substrate - luciferin, which releases light through the bioluminescence process. The desired emission in the range of λ > 600 nm occurs in the reactions that use luciferase from the species *Photinus pyralis* and *Pyrophorus plagiophthalamus* [78]. The maximum bioluminescence effect can be detected 10 min after D-luciferin treatment and lasts up to 30 min. D-luciferin is a relatively stable substrate in vivo. However, the use of luciferase has numerous disadvantages [79,80,81]. The signal of light emitted may be changed by different parameters such as oxygen content and haemoglobin. Numerous variances make it difficult for scientists to analyse experiments, even if performed with the same configuration.

GFP is a widely used reporter gene, recovered from jellyfish *Aequorea victoria* [82] (Figure 2). 

GFP is an excellent indicator of gene transfection and expression in the muscles [83], as the green fluorescence of muscle fibres can be observed through the skin on the same animal by midpoints of time-lapse fluorescence imaging. GFP is already detected 24 h after electrotransfer and remains visible until day 72. The GFP fluorescence should be assessed on a daily and then weekly basis until the green GFP fluorescence is no longer measurable. The main advantage of GFP protein is its intracellular accumulation, which allows direct observation of fluorescence in living cells over time. However, GFP emits fluorescence at wavelengths close to the green spectrum that are absorbed by tissues and additionally, autofluorescence of tissues interferes with fluorescent signals in the green spectral range. Taking all of the above, GFP has all the relevant properties of a quantitative reporter protein. 

The next frequently used reference gene is bacterial beta-galactosidase [7,8,57]. Similar to luciferase and GFP, beta-galactosidase also allows the detection of gene expression in muscle cells after electroporation. Beta-galactosidase cleaves the X-Gal compound and produces an intense blue product. The blue fluorescence can be detected in frozen sections or on fixed tissue. Unfortunately, false positive staining may appear. Furthermore, X-Gal histochemistry after gene transfer of beta-galactosidase coding constructs may underestimate the anatomical range of gene expression [84].

## 5. Skeletal Muscle as A Target for Electroporation 

Skeletal muscles, due to their anatomical properties, are an attractive target in gene therapy, especially in the purveyance of systemic therapeutic proteins. A comparable level of production would depend on the type of muscle selected for electrotransfection (mainly slow (type I), such as soleus, or fast (type II), such as gastrocnemius in mice), fat and fibre content, age of the animal and species used [85,86]. It is worth mentioning that the muscle mass constitutes almost 40% of body weight. Additionally, the extensive network of capillaries surrounding the myotubules promotes the ability to distribute newly synthesized therapeutic agents in the bloodstream. Expression of the transgene persists in the muscles for a long time. First, the nuclei of muscle fibre cells are post-mitotic and terminally diversified. Furthermore, muscle fibres when are locally damaged do not degenerate entirely. That ensures the continuous production of the transgene [87]. Muscle injury after electrotransfection assessed by the degree of infiltration of inflammatory mononuclear cells and necrosis of muscle fibres, reveal decreased efficiency of transfection by a reduced number of muscle fibres capable of expressing the transgene [88]. Myocytes owe a great regeneration ability to satellite cells, located in the niche between the muscle fibre membrane and the underlying basal membrane. Satellite cells are the source of myoblasts necessary for the growth and regeneration of skeletal muscles. Besides the ability to transform into myoblasts, satellite cells also preserve the ability for population renewal, thanks to which they meet the criteria for tissue-oriented targeted, unipotential stem cells. Typically, the satellite cells remain in a resting state until the moment of muscle damage, when muscle regeneration and recovery begins [89,90]. Additionally, the regeneration of muscle fibres from transfected myogenic satellite cells may be the primary mechanism for prolonged transgene expression [91]. Individual skeletal muscle cells are syncytial cells (have many cell nuclei), which ensures efficient distribution of the transgene to adjacent cell nuclei within the myocyte.

## 6. Hyaluronidase Pre-Treatment Improves Vectors Distribution within Muscles

The efficiency of gene transfer in skeletal muscle depends mainly on the widest possible distribution of DNA in the area to which electrical pulses are applied. However, the diverse structure of the connective tissue that tightly covers the entire surface of myofiber and muscle fibre bundles limit the distribution of the injected plasmid, thereby reduce the area of fibres transfection. Also, the size of the pores in the basal lamina selectively limit the penetration of larger viruses, such as adenovirus, compared to smaller adeno-associated viruses (AAV) [92]. Thus, various proteases affecting the muscle extracellular matrix permeability were tested to improve the distribution of both viral and non-viral vectors. Hartikka et al. demonstrated that the injection of the non-ionic surfactant poloxamer 188 (p188), reduces the electroporation-related increase in serum creatine phosphokinase activity and independently increases the expression of the transgene [93]. Favre et al. used hyaluronidase, which hydrolyses hyaluronic acid—the main component of the extracellular matrix of the muscles—reporting even a threefold increase in the efficiency of distribution of adenovirus after injection [94]. Thus, the initial intramuscular administration of hyaluronidase improves both the penetration of the naked plasmid within the tissue and increases its expression but hyaluronidase pre-treatment also reduced muscle damage [33]. 

The short-term toxicity of 100 units of hyaluronidase was tested on two rats sacrificed 3 h following enzyme injection. Histological examinations performed by haematoxylin and eosin staining on muscle tissue did not show differences between tissue after hyaluronidase injection compared to saline-injected rats. Likewise, staining of the basal membrane showed that hyaluronidase does not adversely affect muscle structure. Long-term toxicity performed on animals following hyaluronidase injection showed the same result [94].

## 7. Procedure of Electroporation

1. At least one day before electroporation remove the fur along the muscle to be transfected. An electric razor and then a hair removal cream can be used and left for 30 s as it can cause skin irritation. After cream removal, rinse the skin with plenty of saline. Then use a soothing cream with allantoin.

2. As an additional precaution, add to the drinking water ibuprofen for one day before and after electroporation. 0.2 mg/mL of ibuprofen in drinking water provide a daily dose of approximately 40 mg/kg ibuprofen in C57BL/6J mice (weighing 25 g) with average daily water consumption of 5–6 mL [95,96].

3. For an anaesthesia procedure use an induction chamber. Start general anaesthesia setting 2–3% isoflurane in oxygen. Then, maintain general anaesthesia using a suitable mask with the concentration of 1.5–2% isoflurane in oxygen for the entire procedure. It is relevant to promptly transfer the animals from the induction chamber under the mask because the animals quickly recover from anaesthesia. Isoflurane is a safe, non-flammable anaesthetic for frequent use. Monitor the anaesthetic depth by toe pinch reflex.

4. Intramuscular administration. Wipe the skin centrifugally with a swab. Spray the skin with a disinfectant from a distance of 20 cm and allow it to dry. Depending on the electroporated muscle, inject hyaluronidase two hours before plasmid introduction (Table 3). 

The plasmid DNA amount is one of the parameters having significant transfection efficiency. The standard volume of injection is 50 μL, although depending on the muscle, the volumes vary from 10–100 μL (the smaller volume, the easier it is absorbed). For muscle, the DNA concentration should be 0.5–1.0 μg/μL. Insert the needle at an angle of 90° in relation to the skin and stretched the skin between the thumb and forefinger (that allows to obtain the shortest path to the muscle). The plasmid suspension should be administered very slowly so that it does not cause tissue disruption during injection and does not subject the animal to unnecessary pain. A too rapid injection could give false-positive muscle fibres. PBS should be used to avoid osmotic shock caused by the injection of the plasmid solution. The injection needle must be parallel to the fibres. Several muscles with electroporation in vivo have been successfully used, including tibialis, soleus, gastrocnemius and femoral extensor [97].

5. Placement of electrodes. The conductive paste is crucial to ensure good electrical contact with the skin. The paste layer between the two electrodes should be thin. Next, place the electrodes around the injection site. There are two types of electrodes penetrating and not penetrating can be used. Penetrating electrodes in the form of two needles are placed parallel to the muscle fibres between the injection site. Typically, electrodes with a length of 5 mm are used to electroporate muscles in rodents, while the distance of 5 mm separating them work properly. There are many non-penetrating electrodes available on the market. These are usually two parallel plates of different sizes with different distances between the electrodes that are fixed or adjustable. Two plates should be located on both sides of the injection. Most protocols recommend applying electrical current just after plasmid DNA injection. Tevz et al. [98] observed almost 50% better transfection efficiency when electrical pulses were applied 5 s after plasmid DNA injection, compared to a time lag ranging from 1 min to 2 h. Probably, the clearance and degradation of the injected DNA is a fast process that proceeds within 1 min. Thus, more than half of the injected DNA is lost [99].

6. To administer electric pulses the square wave pulse generator is needed. The number of applied pulses, pulse width and used voltage will vary depending on the tissue and the type of electrode utilized (Table 3). When delivering the transgene, it is worth considering a change in the orientation of the pulses and the location of the electrodes, which improve the effectiveness of the electrotransfer. Applying an electric charge between parallel metal plates, causes the electric current to have a constant value and direction. Hence, DNA migration occurs only on one pole of permeabilized cells [100]. As the tissue conductance is affected by the field-induced cell membrane permeabilization, the time constant of a capacitor discharge pulse generator is changing during the pulse. That leads to a loss in control in the duration of the useful pulse. Thus, there is a need to control pulse delivery. Because of the high current, the voltage is slightly decreasing at the end of the millisecond pulse. Regard to the pulse generator limitation, the work with a frequency greater than 1 Hz is difficult.

7. Take the animal from the anaesthetic mask and let it recover in a separate cage.

8. After finishing the procedures, clean the surface of the electrodes carefully to prevent them from rusting due to electrochemical reactions during the delivery of impulses through the conductive gel layer.

### 7.1. Critical Parameters of Electroporation Depending Muscles Used

#### 7.1.1. Gastrocnemious Muscle 

Two hours before electroporation inject 30 μL of hyaluronidase (0.4 U/μL hyaluronidase in sterile Tyrode) in two divided doses of 12.5 μL from two sides of the muscle. Five seconds after the plasmid injection at a volume of 50 μL, apply on each side of the leg percutaneous electrical impulses through two 10 × 10 mm stainless steel electrodes. Previously, apply a small amount of conductive gel to the electrode plates to ensure good electrical contact and so that the gel layer did not extend beyond the electrodes. The distance between the electrodes is usually about 5–6 mm. Change the position of electrodes by rotation around the muscle. Rotation of 90° ensures the orientation of the field in a perpendicular direction. Place electrodes to the muscle until complete contact with the skin but without muscle compression. Using an electroporator deliver eight, 100-ms/each pulses at a frequency of 1 Hz and 175 V/cm current [101]. It is worth noting that the current voltage of 175 V/cm used in the protocol requires a 105 V voltage setting in the electroporator, which corresponds to a distance of 6 mm between the electrodes (voltage/distance). The intensity of the electric field but not the voltage itself, is directly related to the effectiveness of electropermeabilization. Therefore, if the electrodes have a distance of 10 mm instead of 5 mm, the voltage of the electrical impulses must be doubled to obtain the same electric field strength. In conclusion, at a distance of 4 mm between the electrodes, the set voltage should be 70 V and in the case of 5 mm 88 V and so on. Therefore, it is recommended to set the voltage of the electroporator each time after measuring the distance between the electrodes. The injection volume also limits the amount of plasmid delivered into the mouse gastrocnemius muscle without overflow. Maximum volume 50 μL and the highest concentrations of available plasmid portions (1 μg/1 μL) [101] should be given in 4 punctures of 12.5 μL according to a two-piece administration regimen, on both sides of the upper and lower part of the muscle. The plasmid muscle injection volume cannot be easily increased for technical reasons. Bettan et al. showed that increasing the dose from 30 μg to 300 μg by repeated injections resulted in 7- to 10-fold higher plasma concentrations [102] (Figure 3).

#### 7.1.2. Tibialis

Slowly inject the 30 μL of 8 U/20μL hyaluronidase and 2 h [103] later inject plasmid suspension in PBS in a volume of 30 μL [104]. Approximately 5 s after the injection, attach around the leg the parallel electrodes covered in a conductive gel to ensure reliable electrical contact. A constant distance of 6 mm between the electrodes allows for good contact with the skin surface. Apply five, 120 V electrical impulses, lasting 20 ms/each at 1 Hz [59,105]. 

#### 7.1.3. Soleus

Before an electrotransfection, make an incision in the skin of the leg and fascia over the muscle and reveal the soleus muscle. Following, inject 10 µl of plasmid DNA suspension. Sew the skin incision using polyamide threads. Then apply the conductive gel over the leg skin to ensure good contact between electrodes and the target tissue during the muscle electroporation. Place two stainless steel electrodes over the skin on both sides of the gastrocnemius (soleus anatomical position is deeper underneath lateral gastrocnemius). Apply eight square-wave pulses, with a length of 20 ms and a voltage of 200 V/cm at 1 Hz frequency. Nevertheless, another approach to increase transfection efficiency is to treat soleus muscle with low voltage pulses (50 V/cm) using spatula electrodes that are placed directly over the exposed muscle [106]. 

#### 7.1.4. Foot Muscles: Flexor Digitorum Brevis (FDB) and Interosseus (IO) Muscles

Slowly inject the 10 μL of hyaluronidase at concentration 2 mg/mL. After one hour, inject 20 μL plasmid suspension [104]. 15 min after the injection, attach the needle electrodes under the skin at the heel and a second one at the base of the toes. Electrodes should be oriented parallel to each other and perpendicular to the long axis of the foot. Apply 20 pulses, 100 V/cm, 20 ms in duration/each, at 1 Hz [107]. 

NOTE: Most publications with plasmid DNA electrotransfer relate to the gastrocnemius muscle or tibialis anterior muscle [59,105]. The relatively high mass of these muscles complicates the evaluation of the entire muscle. In addition, the anterior tibialis does not have tendon on the proximal side making it difficult to connect to the dynamometer and evaluate its contractility in vitro. In contrast, the soleus muscle of an adult laboratory mouse has a small mass (~ 9 mg) and tendon at both ends [108]. Muscles differ in the type of fibres the tibial muscle mainly has type IIB and IIDB fibres [109] absent in human muscles, whereas soleus contains part of type I, IIA, IIAD and IID fibres, with a domination of type I and IIA fibres [110,111], which is comparable to human locomotive muscles. Soleus which is a slow contracting muscle does not have type IIB and IIDB fibres. Type IIB and IIDB fibres are the most abundant in gastrocnemius muscle, followed by type IIAD, I, IIA and IID fibres. On the other hand, the tibialis and gastrocnemius muscles are predominantly fast contracting muscles with a larger number of type IIB and IID fibres [112,113].

## 8. Toxicity

In several studies that aimed to optimize the parameters of electroporation, both the significant variability of transgene expression and the degree of tissue damage were observed [88,101,114]. Lefesvre et al. performed histological studies and marked the level of creatine kinase in plasma after electroporation of the gastrocnemius muscle [101]. Histological examinations showed muscle necrosis with a strong mast cell and eosinophils infiltration, with more severe damage observed between days 7 and 14, indicating slow necrosis of muscle fibres in the first days after electroporation. Hartikka et al. showed that within the infiltrating cell population was mainly mononuclear cells but also leukocytes and polymorphonuclear lymphocytes [93]. Additionally, some researchers believe that infiltration of inflammatory mononuclear cells after electrotransfer stimulates the immune response to vaccines introduced by electroporation [98]. Intramuscular injections of physiological saline significantly increased muscular injuries caused by the applied electrical current. However, there was no further rhabdomyolysis after using the suspension of plasmids. The degree of necrosis correlates with the applied voltage above 100 V/cm. Notably, muscle damage is not affected by the use of simple anaesthesia in combination with muscle relaxants, different leg positions between the electrodes nor mouse sex. Gehl et al. demonstrated that the use of a needle or plate electrode during electroporation does not significantly affect muscle damage [115] and the toxicity itself was mainly related to the degree of permeabilization of cells [116]. However, the observed changes were transient. 28 days after electroporation, regions containing necrotic muscle fibres and mononuclear infiltrates were regenerated. There were visible mineralized remnants of muscle fibres suggesting scar tissue. After 56 days there was no histological difference between control muscles and those after electroporation [93]. 

Mathiesen described the necrosis of the electroporated muscle with the pre-injection of the plasmid suspension along with the cumulative duration of the pulses [57]. There was no expression of the reporter gene in damaged muscle fibres. It is important to emphasize that after muscle electroporation, there was no expression of the transgene or report protein in the liver. 

## 9. Effectiveness of Electroporation 

The efficiency of electroporation depends on the composition of the cell membrane, the size and conformation of the DNA molecule, the duration of the individual pulses, as well as the voltage applied to the given tissue. Bettan et al. [102] and Mir et al. [7] set electroporation parameters (8 pulses, 20 ms and 200 V/cm at 1 Hz), which guaranteed high plasmid expression and the ability to determine proteins in the blood. An important issue is the quantitative evaluation of the effectiveness of electroporation and expression of the delivered genes to target tissues in live animals. In vivo, optical imaging is a non-invasive method that can identify and track the activity of the delivered gene on the live animal at various time points [117], what both reduces the number of laboratory animals and improves the efficiency of statistical analysis.

Expression of the protein after electroporation can be tested 24 h after transfection of the plasmid for up to 6 months [107]. In the absence of an immune response, there was no reduction in the number of transfected fibres up to a year after injection [85]. Depending on the electroporated tissue and the type of plasmid, protein expression can be determined depending on whether it is secreted into the bloodstream or whether it only affects the injection site. In the first case, the expression sequence at different time points can be easily counted by blood sample collection. In the second, the level of the protein can be determined after homogenization of the proper tissue. An excellent solution is an electroporation with the use of a plasmid which contains both the proper silencing gene and a reference gene, for example, green fluorescent protein (GFP) which is easily determined by in vivo imaging (Figure 4). 

If a depilatory cream is used to remove fur, it is relevant to perform the shaving activity at least 24 h before fluorescent imaging, because some crème ingredients fluoresce when the blue excitation light is used. Introduction of plasmid DNA intramuscularly in pigs caused the appearance of pDNA in the bloodstream but at a relatively low level, about 10% of all administered pDNA, with the highest concentration 15 min after administration and then drastically falling [118]. It was also shown that the plasmid DNA found in the bloodstream is degraded rapidly by nucleases in the blood: 20.9% after 10 min of intramuscular injection, 34% after one hour, 86.6% after one day and almost complete removal (97.8%) after one week [119].

## 10. Limitations of Electroporation

Despite the advantages of electroporation, the in vivo method cannot be widely used due to some limitations. Significant is the small range between the electrodes, usually around 1 cm, which limits the transfer of genes to a large area of tissues. Conduction of the electrotransfer in internal organs is problematic in view of the need for surgical procedure and the fear of organ damage by the use of high voltage [120]. Electroporation can cause severe collateral damage. When the electric field is applied using plate electrodes, the major potential decline occurs over the skin instead of the targeted subcutaneous tissues. Skin oedema is a frequent consequence. A few hours after electroporation, we also observed redness and swelling at the place where the electrodes were applied. Before attempting to supply plasmid DNA to the human body, it would be useful to perform similar tests for a larger primate to develop an optimal plasmid introduction method [121]. Roche et al. noted that the use of electroporation parameters (5 impulses at an intensity of 180 V/cm, at 20-ms duration per pulse, separated by a 200-ms interpulse interval) necessary for the transfection of about 25% of muscle fibres in tibialis is harmful, and regeneration requires myogenesis, which can last for many weeks. Additionally, they described a decrease in muscle contraction by about 80% just after electroporation. Muscles reached 40% of their contractile function within 3 h after delivery of the electrical impulse [122]. Nevertheless, non-viral vectors intended for commercial and clinical use must be well characterized in terms of scaling up the production process and long-term storage. The question of the persistence and accumulation of biomaterials in the body is still an issue that is poorly understood. It is postulated that such a situation may induce inflammation or even be toxic. 

## 11. Discussion

Attempts to develop a manageable, potent and cost-effective protocol of an in vivo gene delivery method of limited tissue damage which will lead to transgene expression, seem to have therapeutic potential in the future. Our review of several electroporation protocols gives us practical information about appropriate method parameters that can be useful in future protocol improvements. Electroporation is an efficient method for long-term expression of in vivo plasmid DNA delivery in various types of tissues and probably to all cell types. The method ensures, in addition to several folds of transgene expression, limitation in individual variability [7]. The first reports on the possibility of using plasmid DNA in gene therapies date back to 1990 when Wolff et al. demonstrated that intramuscular administration of plasmid DNA caused expression of the transgene [61]. Various factors for example, acute inflammatory reactions to vectors used in gene therapies, carcinogenesis or limited therapeutic effect caused the slowdown in the development of effective gene therapies [123]. Currently, there are a number of disease entities for which clinical trials are conducted using plasmid DNA based gene therapy for example, cancer (malignant melanoma, pancreatic cancer, breast cancer) [124], heart disease (ischemic limb disease, coronary artery disease), muscular dystrophy [4], infectious diseases (HIV, hepatitis B), hereditary monogenic diseases (Huntington’s chorea, haemophilia) [75]. Additionally, the combination of plasmid DNA vaccination and electroporation appears to be a promising strategy in the prevention of infectious diseases and cancer immunotherapy. DNA vaccines are cost-effective, manageable and considered harmless. The adjuvant electroporation properties are mainly due to enhance the “danger signal” that become recognized by the immune system. The tissue damage caused by the electric field recruits macrophages, lymphocytes, dendritic and T-cells to the injection site provoking increased immune responses [54,55,56]. 

Electrotransfer was used in various muscles of different species, including primate, which indicates a wide range of applications [7]. The electroporation uses the fact that the cell membrane acts as an electric condenser which, with the exception of the ionic channels, is unable to pass the current. An electric high voltage field applied to the membranes causes the temporary formation of pores large enough to allow the macromolecules to enter or leave the cell. Re-closing the pores of the membrane is a natural process. At a time when the pores are open, the nucleic acid can get into the cell and eventually into the nucleus. The degree of muscle damage depends on the parameters of the electrical pulses used [37,57,60], as well as on the design of the electrode itself [117]. Mir et al. conducted the electrotransfer on four animal species, including monkeys and showed that the use of external plate electrodes gives a similar expression of the transgene to those obtained with invasive needle electrodes [7]. Thus, appropriately selected electrical impulses increase the gene transfer both by the formation of pores in the cell membrane and by direct effects on the DNA molecule. However, several protocols of transgene electrotransfer suggest that limited damage of muscle tissue caused by an electric field near the electrodes may be unavoidable [117]. 

Intramuscular injection of the plasmid is a potential alternative for viral vectors to transfer therapeutic genes to skeletal muscle fibres. Low efficiency of plasmid-based gene transfer is enhanced by electroporation coupled to the intramuscular use of hyaluronidase [34]. For example, using typical electroporation parameters, up to 22% of transfection efficiency can be obtained in the tibialis anterior muscle. Increasing efficiency by using higher voltage or increasing the amount and time of impulse will be associated with significant damage to the muscle and skin burns. Molnar et al. showed that hyaluronidase administration before electroporation increases the transfection efficiency in the anterior tibialis muscle [85].

The use of appropriate parameters during the electrotransfer of plasmids to the muscles ensures long-term expression of the transgene. The main advantages of muscle electroporation refer to their further exogenous action, that is, the potential for the production and secretion into the bloodstream of bioactive proteins that can have an effect on distant tissues such factors as interleukin-5 [8], erythropoietin [52,125,126], fibroblast growth factor 1 [7], interferon-α [127] and human factor IX [128]. The use of electroporation for intramuscular gene delivery is particularly relevant for vaccination purposes [3] and the delivery of anticancer vaccines [129]. Initially, the research used either high voltage and short pulses or low voltage and long pulses electroporation protocols. The highest levels of the encoded protein in the plasma were obtained at low voltages (200 V/cm) of long pulses (20 ms) and 8 pulses at 1 Hz. The blood concentration of the human secreted alkaline phosphatase reached about 2 μg/mL [102]. However, short (100 ms) and intense (800 to 1500 V/cm) electrical impulses used in electrochemotherapy protocols are safe and well tolerated. The electrochemotherapy used in oncology facilities entry of hydrophilic anti-cancer agents such as bleomycin [130] into the cell, which drastically enhances its anticancer effects. Many reports have shown that increasing the voltage during electroporation increases both the permeability of biological membranes and the efficiency of transfection. However, great efficiency in delivering a transgene is associated with significant muscle damage and reduced muscle contractile strength [88,101,114,122].

The method of plasmid electrotransfer has many advantages over the use of viral vectors, thanks to the simplicity of plasmid production, safety, reduced costs and the possibility of targeted therapy [131]. Plasmids pose natural ability to stay stable in bacterial cells, which promotes their genetic modification. Additionally, depending on the species used for the study, differentiated muscle susceptibility to intramuscular electrotransfer of plasmids can be expected. Hitherto, clinical trials involving people in whom electroporation was used were related to electrochemotherapy of tumours [132]. In 1991, Mir et al. published the first clinical trial of electrochemotherapy with the use of bleomycin, performed at the Gustave-Roussy Institute in eight patients with recurrent or progressive squamous cell carcinoma of the head and neck. To date, several clinical trials have been conducted in various cancers [133]. Although the electrochemotherapy procedure alone did not cause permanent side effects, further research is required to establish safe parameters of the electrotransfer for skeletal muscle. Numerous experts in the gene therapy field consider electroporation extremely promising for biology, medicine, biotechnology, food, environmental technologies, agriculture, process engineering, energy and environment. For this reason, an International Society for Electroporation-Based Technologies and Treatments (ISEBTT) was established and a comprehensive “Handbook of Electroporation” [134] was published to disseminate technology, provide information on electroporation in the scientific community and improve protocols. 

In summary, electroporation is one of the most promising physicochemical methods for gene and drug delivery but it also has several disadvantages. In order to reduce the main side effects of muscle electroporation, that is, muscle contraction and impairment of its contractile function, it is required to establish the appropriate number and quality of impulses in the test protocol and to optimize the structure of the electrodes used. 

## Figures and Tables

**Figure 1 ijms-20-02776-f001:**
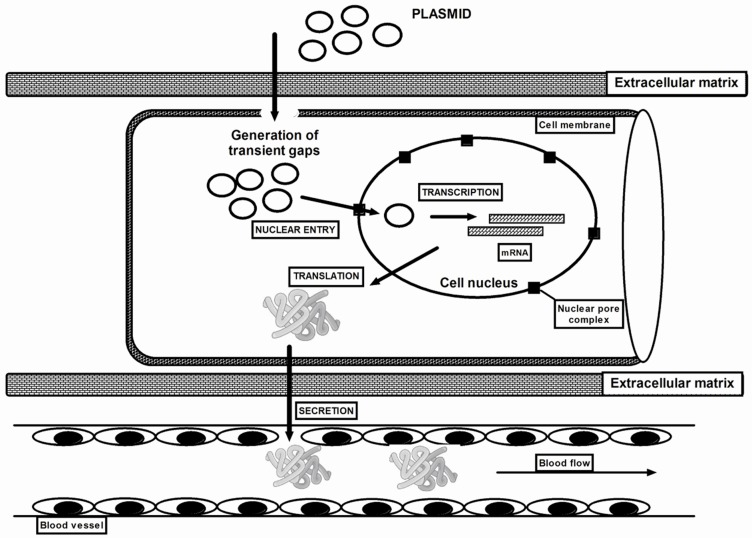
Schematic diagram of proposed plasmid DNA delivery with the use of electroporation. Plasmid DNA (pDNA) must overcome barriers, including extracellular matrix, cell membrane, cytoplasm and nuclear membrane before reaching the nucleus. However, applying an electric current with an appropriate voltage facilitate the transport of pDNA through biological membranes. Plasmids penetrate into the target cell cytoplasm through the transiently formed gaps in the sarcolemma. DNA in the cytoplasm is exposed to degradation by cytoplasmic nucleases. Finally, the DNA through nuclear pore complexes has to penetrate the nuclear membrane to enter the nucleus where the transcription occurs. The delivered plasmid DNA may silence genes or be transcribed into the missing protein and then secreted into the bloodstream.

**Figure 2 ijms-20-02776-f002:**
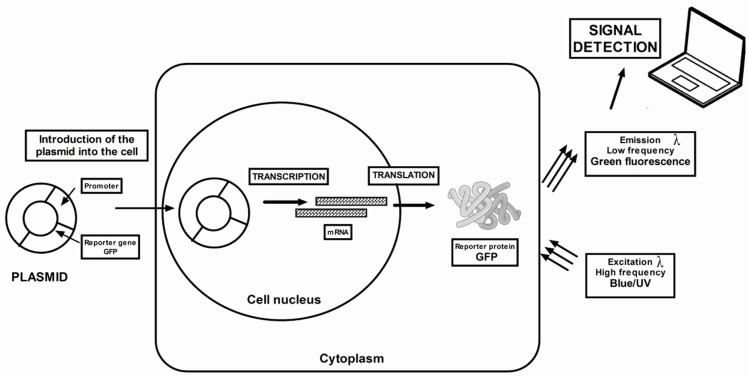
Diagram of the bioluminescence imaging technique using the green fluorescent protein (GFP) reporter gene. The GFP reporter gene is directly attached to the gene of interest to produce gene fusions. The two genes are located under the same promoter elements and are transcribed into a single messenger RNA molecule. The mRNA is then translated into a protein. The protein-GFP hybrid transcribed from the reporter construct possesses a protein linked to GFP. Based on the fluorescence, it can be concluded that the protein is everywhere where green fluorescence occurs.

**Figure 3 ijms-20-02776-f003:**
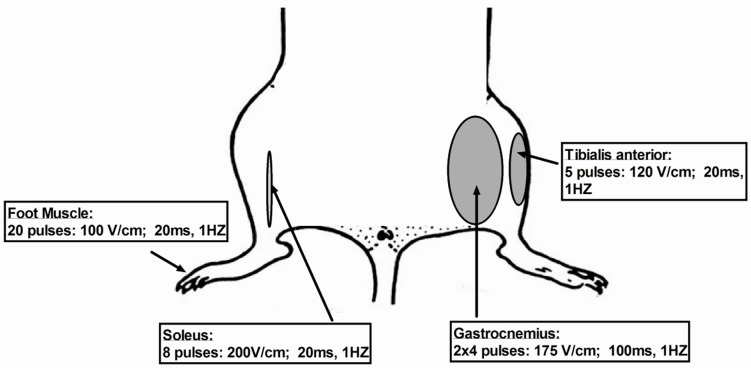
Schematic representation of electroporation parameters.

**Figure 4 ijms-20-02776-f004:**
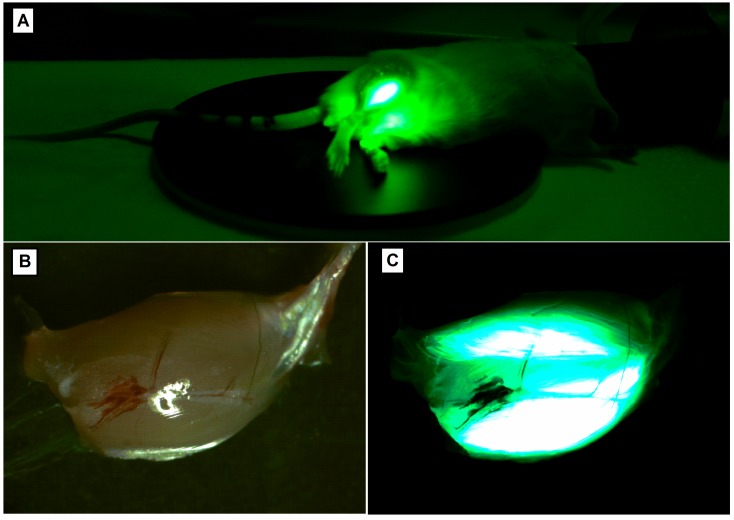
The photographs made by our research team show the bioluminescence of GFP as a reporter protein 3 days after electroporation (**A**) of tibialis muscle in vivo in a living mouse and (**C**) ex vivo in gastrocnemius muscle. The green fluorescence of muscle fibres can be observed through the skin on the same animal by means of time-lapse fluorescence imaging (**A**). GFP emits green light following excitation by wavelength in the UV range. (**B**) shows muscle viewed in the range of visible light.

**Table 1 ijms-20-02776-t001:** Advantages and limitations of the use of viral vectors in gene therapy.

Used Vectors	The Main Advantages	Limitations
Adenoviruses	High efficiency of in vivo and ex vivo transduction; High level of transgene expression; Possibility of obtaining high titre virus preparations.	Cytotoxicity; Strong immune response to viral proteins restricting repeated administration; Short-term transgene expression (no integration with the genome).
Retroviruses	Long-term transgene expression (integration with the genome).	Introduction of the transgene possible only to dividing cells; Possible insertion mutations (integration into the genome); Application limited mainly to transducing cells ex vivo.
Lentiviruses	Introduction of the transgene possible also to non-dividing cells; Long-term transgene expression;	Possible insertion mutations.
AAV (Adeno-associated viruses)	Low immunogenicity; Introduction of the transgene possible also to non-dividing cells; Long-term transgene expression.	Possible insertion mutations; Difficult quality control.
Herpes virus	Introduction of the transgene possible also to non-dividing cells; Transgenes up to 15 kbp.	Short-term expression of the transgene.

**Table 2 ijms-20-02776-t002:** Electroporation and sonoporation as non-viral methods of transfer of genetic material in gene therapies.

Method for Genetic Material Introduction	Method of Application	Advantages	Limitations
Electroporation	Uses an electric current pulse to form a transient branches in the cell membrane	Highly effective, reproducible, directed gene transfer, the possibility of transferring the DNA macromolecule	Impossible to use on a large area; requires surgical intervention when transfer to internal organs; the use of high voltage can disturb the DNA
Sonoporation	Uses ultrasounds to induce a transient state of cell membrane permeability	Harmless, non-invasive, DNA transfer into internal organs without the need for surgical procedure	Low efficiency

**Table 3 ijms-20-02776-t003:** Parameters of electroporation. Depending on the electroporated tissue, the volume and concentration both hyaluronidase and plasmids suspension, as well as the number of applied pulses, pulse width, applied voltage and the type of electrode will differ.

Muscle				
	Gastrocnemious	Tibialis	Soleus	Foot Muscle: (Flexor Digitorum Brevis (FDB) and Interosseus (IO) Muscle)
Hyaluronidase in sterile Tyrode	0.4 U/μL (30 μL)	8 U/20 μL	-	6 U/10 μL
Plasmid volume (μL)	50 (4 × 12.5) (1–2 μg/1 μL)	30 (2 μg/1 μL)	20 (2 μg/1 μL)	2–5 μg plasmid/μL volume (10–20 μL)
Syringe needle	29 G	26 G	34 G	33 G
Pulses	2 × 4	5	8	20
Voltage (V/cm)	175	120	200	100
Pulse width (ms)	100	20	20	20
Frequency (Hz)	1	1	1	1
Type of electrodes	plate electrodes 10 × 10 mm	plate electrodes 7 × 7 mm	plate electrodes 10 × 10 mm	plated acupuncture needle

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
