# Peer review of "A Critical Review of Electroporation as A Plasmid Delivery System in Mouse Skeletal Muscle"

_ijms, 2019, doi:10.3390/ijms20112776_

Round 1
Reviewer 1 Report
In the present study, the authors reviewed the useful technique of electroporation to deliver a plasmid into mouse skeletal muscle and other tissues. The manuscript was well written and organized. There is no major concern but this review recommends authors to write a separate section regarding the current limitations of this technique. For example, the number of muscle fibers that express the plasmid that was injected by the electroporation technique or only local expression will be achieved.
Author Response
Point 1: There is no major concern but this review recommends authors to write a separate section regarding the current limitations of this technique.
Response 1: Thank you for your valuable comments. As suggested, we separated an additional section in the manuscript describing the limitations of electroporation. The introduction of an additional section (see Paragraph 10: Limitation of electroporation, Line: 432-542) significantly improves the organization and readability of the manuscript.
Reviewer 2 Report
The manuscript presents an extensive overview on Electroporation as a methodology for a more efficient plasmid DNA delivery. The authors describe the past and recent progress in gene therapy based on naked DNA, discussing the electroporation as a promising tool for the transfer of nucleic acids. In particular, they focus on intramuscular injection in different muscles, suggesting protocol conditions and parameters following their own research experience and reporting data from the literature.
The paper is very interesting and well written: all the aspects about safety of plasmid DNA as therapeutic agent and its possible applications into the clinics are described; the mechanisms of in-take and failure of DNA, if injected alone, to express a large amount of the protein of interest, are provided to introduce electroporation as a useful delivery strategy. The paper offers explicative figures and tables. The bibliography is extensive.
Due to the important role that Electroporation is covering in the field of immunization, I would suggest to briefly discuss its properties as “adjuvant approach”. Some short concepts could be added in the text, as suggested below.
Due to the growing interest in this field, I would consider this review acceptable for publication, after the minor revisions suggested to the authors.
Minor Revisions:
Line 91: Because electroporation has been proven as a strategy which can enhance an immune response by the recruitment of cells of both innate and adaptive immune system, the authors could report and describe the adjuvant properties of electroporation, discussing its important role in vaccination/immunization protocols, by the help of the following papers
Bachy, M.; Boudet, F.; Bureau, M.; Girerd-Chambaz, Y.; Wils, P.; Scherman, D.; Meric, C. Electric pulses increase the immunogenicity of an influenza DNA vaccine injected intramuscularly in the
mouse. Vaccine, 2001, 19 (13-14), 1688-93.
Babiuk, S.; Baca-Estrada, M. E.; Foldvari, M.; Middleton, D. M.; Rabussay, D.; Widera, G.; Babiuk, L. A. Increased gene expression and inflammatory cell infiltration caused by electroporation are both important for improving the efficacy of DNA vaccines. J. Biotechnol.,
2004, 110(1), 1-10.
Chiarella, P.; Massi, E.; De Robertis, M.; Sibilio, A.; Parrella, P.; Fazio, V. M.; Signori, E. Electroporation of skeletal muscle induces danger signal release and antigen-presenting cell recruitment independently of DNA vaccine administration. Expert Opin. Biol. Ther. 2008, 8(11), 1645-57.
Chiarella P, Fazio VM, Signori E. Electroporation in DNA vaccination protocols against cancer. Curr Drug Metab. 2013 Mar;14(3):291-9
Line 112: please check the cited references 7,8,54: with LV are used ms rather than ms
Line 134: in the Table correct “can disturbed”
Line 259: in the Table correct “pate” with plate
Line 497: this last part of the manuscript sounds as a “short cut” of the paper. I would suggest to improve this part with some conclusive sentences in favor of electroporation (as I intended the aim of this review). In this final part (lines 488-497), the authors seem not so confident in the use of electroporation: they conclude with a list of “deficiencies” rather than listing the promising advantages of this approach. Numerous experts in gene therapy field consider EP so promising and not only for biomedical applications. For this reson, an International Society was recently found (ISEBTT) and an extensive book (that should be cited in the bibliography) has been published with the purpose to spread the technology and the different protocols in the scientific community - Please, refer to this link, if the authors decide to accept this comment and want to cite correctly the book:
https://link.springer.com/referencework/10.1007/978-3-319-26779-1
Author Response
Point 1: Line 91: Because electroporation has been proven as a strategy which can enhance an immune response by the recruitment of cells of both innate and adaptive immune system, the authors could report and describe the adjuvant properties of electroporation, discussing its important role in vaccination/immunization protocols, by the help of the following papers:
Bachy, M.; Boudet, F.; Bureau, M.; Girerd-Chambaz, Y.; Wils, P.; Scherman, D.; Meric, C. Electric pulses increase the immunogenicity of an influenza DNA vaccine injected intramuscularly in the mouse. Vaccine, 2001, 19 (13-14), 1688-93.
Babiuk, S.; Baca-Estrada, M. E.; Foldvari, M.; Middleton, D. M.; Rabussay, D.; Widera, G.; Babiuk, L. A. Increased gene expression and inflammatory cell infiltration caused by electroporation are both important for improving the efficacy of DNA vaccines. J. Biotechnol., 2004, 110(1), 1-10.
Chiarella, P.; Massi, E.; De Robertis, M.; Sibilio, A.; Parrella, P.; Fazio, V. M.; Signori, E. Electroporation of skeletal muscle induces danger signal release and antigen-presenting cell recruitment independently of DNA vaccine administration. Expert Opin. Biol. Ther. 2008, 8(11), 1645-57.
Chiarella P, Fazio VM, Signori E. Electroporation in DNA vaccination protocols against cancer. Curr Drug Metab. 2013 Mar;14(3):291-9
Response 1: Thank you for your valuable comments. As suggested, we highlighted the important role of electroporation in vaccination / immunization protocols (Please find: Line 94-97 and Line 469-475). Thank you for the given literature. Citations number: 55, 56, 57.
Point 2: Line 112: please check the cited references 7,8,54: with LV are used ms rather than ms
Response 2: We've corrected the mistake from μs to ms. Line: 119.
Point 3: Line 134: in the Table correct “can disturbed”
Response 3: We've corrected the mistake from can disturbed to can disturb. Line: 140.
Point 4: Line 259: in the Table correct “pate” with plate
Response 4: We've corrected the mistake from pate to plate. Line: 267.
Point 5:Line 497: this last part of the manuscript sounds as a “short cut” of the paper. I would suggest to improve this part with some conclusive sentences in favor of electroporation (as I intended the aim of this review). In this final part (lines 488-497), the authors seem not so confident in the use of electroporation: they conclude with a list of “deficiencies” rather than listing the promising advantages of this approach. Numerous experts in gene therapy field consider EP so promising and not only for biomedical applications. For this reson, an International Society was recently found (ISEBTT) and an extensive book (that should be cited in the bibliography) has been published with the purpose to spread the technology and the different protocols in the scientific community - Please, refer to this link, if the authors decide to accept this comment and want to cite correctly the book: https://link.springer.com/referencework/10.1007/978-3-319-26779-1
Response 5: Thank you for your valuable comments. As suggested, we emphasized the importance of electroporation in various fields of science (Line: 527-533); we also cited the proposed Handbook (Citations number: 47, 135), which describes in detail the many outlines of electroporation. This is a handbook that I will often use. Thank you.